# Exploratory Analysis of Lenvatinib Therapy in Patients with Unresectable Hepatocellular Carcinoma Who Have Failed Prior PD−1/PD-L1 Checkpoint Blockade

**DOI:** 10.3390/cancers12103048

**Published:** 2020-10-20

**Authors:** Tomoko Aoki, Masatoshi Kudo, Kazuomi Ueshima, Masahiro Morita, Hirokazu Chishina, Masahiro Takita, Satoru Hagiwara, Hiroshi Ida, Yasunori Minami, Masakatsu Tsurusaki, Naoshi Nishida

**Affiliations:** 1Department of Gastroenterology and Hepatology, Kindai University Faculty of Medicine, 377-2 Ohno-higashi, Osaka-Sayama, Osaka 589-8511, Japan; t.aoki1918@gmail.com (T.A.); kaz-ues@med.kindai.ac.jp (K.U.); s0750081@yahoo.co.jp (M.M.); chishina@med.kindai.ac.jp (H.C.); masahirot2797@yahoo.co.jp (M.T.); hagi-318@hotmail.co.jp (S.H.); hidakuhp@gmail.com (H.I.); minkun@med.kindai.ac.jp (Y.M.); naoshi@med.kindai.ac.jp (N.N.); 2Department of Radiology, Kindai University Faculty of Medicine, 377-2 Ohno-higashi, Osaka-Sayama, Osaka 589-8511, Japan; mtsuru@dk2.so-net.ne.jp

**Keywords:** hepatocellular carcinoma, lenvatinib, PD-1/PD-L1 blockade, molecular targeted agents, vascular endothelial growth factor

## Abstract

**Simple Summary:**

Programmed cell death protein 1 (PD−1)/PD-ligand 1 (PD-L1) blockade is becoming a novel therapeutic option for a hepatocellular carcinoma. In this work, we evaluated efficacy and safety of lenvatinib following failure of PD-1/PD-L1 blockade. The median progression-free survival was 10 months (95% confidence interval (CI): 8.3–11.8) and the median overall survival was 15.8 months (95% CI: 8.5–23.2) since lenvatinib therapy initiation. The objective response rate was 55.6%, and the disease control rate was 86.1%. All of efficacy outcomes were better than those by lenvatinib treatment alone as the 1st line treatment therapy. No particular safety concerns were observed. It was speculated that lenvatinib right after failure of PD-1/PD-L1 blockade provided synergistic effect since anti-PD-1 antibodies can remain binding to CD8+T cells for more than several months. Lenvatinib demonstrated considerably high antitumor activity and good survival benefit with acceptable toxicity in patients with unresectable HCC when administered right after failure of PD-1/PD-L1 blockade.

**Abstract:**

Although programmed cell death protein 1 (PD−1)/PD-ligand 1 (PD-L1) blockade is effective in a subset of patients with hepatocellular carcinoma (HCC), its therapeutic response is still unsatisfactory. Alternatively, the potential impact of the lenvatinib in patients who showed tumor progression on PD−1/PD-L1 blockade is unknown. In this work, we evaluated the safety and efficacy of lenvatinib administration after PD-1/PD-L1 checkpoint blockade. The outcome and safety of lenvatinib administered after PD-1/PD-L1 blockade failure was analyzed retrospectively in 36 patients. Tumor growth was assessed every 4–8 weeks using modified Response Evaluation Criteria in Solid Tumors. The mean relative dose intensity of lenvatinib was 87.6% and 77.8% in patients receiving a starting dose of 8 (interquartile range (IQR), 77.5–100.0) mg and 12 (IQR, 64.4–100.0) mg, respectively. Since lenvatinib therapy initiation, the median progression-free survival was 10 months (95% confidence interval (CI): 8.3–11.8) and the median overall survival was 15.8 months (95% CI: 8.5–23.2). The objective response rate was 55.6%, and the disease control rate was 86.1%. No particular safety concerns were observed. Lenvatinib demonstrated considerable antitumor effects with acceptable safety in patients with progressive and unresectable HCC when administered right after PD-1/PD-L1 blockade failure.

## 1. Introduction

Hepatocellular carcinoma (HCC) is the most common primary malignancy of the liver and one of the major causes of cancer-related death worldwide [1,2,3,4,5]. Unfortunately, owing to the late occurrence of symptoms, HCC progression is often diagnosed at an advanced state, which prevents the application of locoregional therapies. Even when patients diagnosed with early or intermediate tumor stages are treated, prognosis is poor because HCC has a high recurrence rate. Considering the high number of HCC cases with progressive disease (PD), the development of an effective systemic therapy has been an urgent need over the past decade for the management of unresectable tumors.

Recently, targeting unique signaling pathways critical in tumor growth has emerged as a promising strategy for HCC treatment. In this context, several molecular targeted agents (MTAs), including sorafenib [6,7], lenvatinib [8], regorafenib [9], ramucirumab [10], and cabozantinib [11], have been used in clinical settings [12,13]. Notably, lenvatinib is a multikinase inhibitor that targets vascular endothelial growth factor (VEGF) receptors 1–3, platelet-derived growth factor (PDGF) receptor alpha, fibroblast growth factor (FGF) receptors 1–4, rearranged during transfection (RET), and KIT, which draws a strong anti-tumor response in HCC cases [14,15,16,17].

Alternatively, immunotherapy is becoming a novel therapeutic option for a variety of cancers [18,19,20]. Antibodies against programmed cell death protein 1 (PD-1) have shown prolonged antitumor responses in patients with advanced HCC [21,22]. For this reason, the Food and Drug Administration (www.fda.gov) approved the anti-PD-1 antibodies nivolumab [23] and pembrolizumab [24] as second-line therapies following sorafenib treatment. However, phase III clinical trials failed to show the survival benefits of these agents for the treatment of unresectable HCC in both first- and second-line settings [25,26]. On the other hands, in patients with unresectable HCC, atezolizumab combined with bevacizumab resulted in better overall and progression-free survival outcomes than sorafenib [27], so combination therapy of anti-VEGF/MTA and anti-PD-1/PD-L1 blockade might be new first-line treatment option for patients with unresectable HCC instead of sorafenib or lenvatinib.

Although a precise mechanism underlying the refractoriness to immune checkpoint inhibitors (ICIs) in patients with HCC remains unexplored, hormones, such as vascular endothelial growth factor (VEGF), could mediate such refractoriness. VEGF secreted by HCC cells could result in ICI resistance by inducing CD8^+^ T cell exhaustion and an immunosuppressive cellular phenotype in the tumor microenvironment (TME) [28,29].It has also been reported that compared with HCC without activating mutations the Wnt/β-catenin pathway, HCC with such mutations can induce an alteration in immune cell recruitment and result in poorer prognosis following ICI treatment. This suggests a role of the Wnt/β-catenin pathway in the establishment of “immune cold” phenotype in HCC [30].

In contrast, considering the potential anti-VEGF activity of MTAs, these agents could alter the immunological microenvironment and reverse the refractoriness to ICIs. In addition, few MTAs exert their antitumor effect through a VEGF-independent mechanism by inhibiting the signaling pathways involved in tumor growth. In this scenario, the combination of ICIs and MTAs could be an attractive option for the treatment of advanced HCC cases and even in those refractory to ICI monotherapies [31]. Indeed, the efficacy of atezolizumab and bevacizumab [27] or pembrolizumab and lenvatinib [32] combination has been confirmed, although no real-world data are available for such combinations for HCC treatment. In contrast, as antibodies are considerably stable in the living body [33], it is reasonable to expect that sequential therapy with MTAs following anti-PD-1 antibody will exert a synergic effect similar to their combinatorial therapy. Among the MTAs available for HCC, lenvatinib showed the most potent antitumor activity [34], and it would be intriguing to evaluate its efficacy in HCC refractory to ICI monotherapy.

Thus, we present here the first real-world data analysis of lenvatinib efficacy and safety following the clinical trial of PD-1/PD-L1 blockade in patients with HCC unresponsive to ICIs. Although this study is conducted as a sequential therapy not a combination, anti-PD-1 antibody is known to bind on CD8^+^ T cells for more than 20 months, where sustained blockade to PD-1 can be expected [32]. In addition, inhibitory activity of lenvatinib against the FGF signaling pathway of HCC is prominent, considering the low Median Inhibition Concentration (IC_50_) value for fibroblast growth factor receptor 4 (FGFR4) [35]. From this point of view, lenvatinib could be effective for HCC with high FGFR4 expression that prone to carry Wnt/β-Catenin mutation [36]. Collectively, it is conceivable that ICI-lenvatinib sequential therapy would be also effective, similar with their combination, which is the clinical question to be clarified in this exploratory analysis.

## 2. Results

### 2.1. Patient Characteristics

A total of 170 patients received lenvatinib at our institution. Between March 2016 and September 2019, 36 of these patients who received lenvatinib as a next-line treatment immediately after anti-PD-1/PD-L1 therapy failure were enrolled in the present analysis. Of these 36 patients, 14 received a systemic therapy before ICI therapy. Patient characteristics have been summarized in Table 1. The median age of patients was 70 (range, 60.8–76.8) years, and most patients were male (31.86%), with a male to female ratio of 6:1. In this cohort, 55.6% (20/36) of the patients were negative for hepatitis B surface antigen (HBsAg) and hepatitis C antibody (HCVAb), 27.8% (10/36) were positive HCVAb, and 16.7% (6/36) were positive for HBsAg. All patients had Child-Pugh grade A liver cirrhosis. The tumors were large and/or multi-nodular in all patients, where 25% of the patients (9/36) were Barcelona Clinic Liver Cancer (BCLC)-stage B [37] and 75% (27/36) of the patients were BCLC-stage C. Vascular invasion and extrahepatic spread were observed in 13.9% and 72.2% patients, respectively. The median α-fetoprotein (AFP) level was 76.5 (interquartile range (IQR), 5.25–11,543.5) ng/mL, with 44.4% of patients having an AFP higher than 400 ng/mL. The median des-γ-carboxy prothrombin (DCP) level was 1303.5 (IQR, 43.25–7394.3) milli arbitrary units per milliliter (mAU/mL), with 63.9% of the patients having a DCP higher than 100 mAU/mL. The duration of prior PD-1/PD-L1 checkpoint blockade ranged from 1.7 to 8.8 (median, 3.7) months.

### 2.2. Administration of Lenvatinib and Outcomes

At the end of the study period (29 February 2020), the median follow-up duration was 5.5 (range, 1.1–20) months. During the observation period, 11 patients died because of HCC progression. The mean duration between ICI therapy termination and lenvatinib treatment initiation was 3.65 (IQR, 0.50–4.46) months, with a median treatment period of 4.6 (IQR, 2.2–10.0) months. The mean relative dose intensity for lenvatinib with starting doses of 8 mg and 12 mg was 87.6% (IQR, 77.5–100.0%) and 77.8% (IQR, 64.4–100.0%), respectively. The median time for lenvatinib first dose reduction was 6.3 (IQR, 2.4–26.0) weeks. Treatment was discontinued in 15 patients owing to disease progression (*n* = 10, 27.8%) or treatment-related adverse events (AEs) (*n* = 5, 13.9%). Fourteen (39%) of the 36 patients with a response are under the ongoing responses.

### 2.3. Antitumor Response and Patient Survival After Lenvatinib Administration Following Immune Checkpoint Inhibitor Therapy

The median progression-free survival (PFS) was 10.0 months (95% confidence interval (CI): 8.3–11.8, Figure 1), and the median overall survival (OS) calculated since the initiation of lenvatinib therapy was 15.8 months (95% CI: 8.5–23.2, Figure 2). The median OS since ICI therapy initiation was 29.8 months (95% CI: 25.3–34.4). The objective response rate (ORR) and disease control rate (DCR) in patients who underwent radiological evaluation were 55.6% and 86.1%, respectively, as assessed using the modified Response Evaluation Criteria in Solid Tumors (mRECIST) [39]. Complete response (CR), partial response (PR), stable disease (SD), and PD were noted in 2.8% (*n* = 1), 52.8% (*n* = 19), 30.6% (*n* = 11), and 11.1% (*n* = 4) patients, respectively. The ORR and DCR, evaluated using RECISTv1.1, were 22.2% (*n* = 8) and 94.4% (*n* = 34), respectively, whereas CR, PR, SD, PD were 0.0% (*n* = 0), 22.2% (*n* = 8), 72.2% (*n* = 26), and 0.0%, respectively.

Early responses of lenvatinib were observed during the first 4 weeks in 30 out of 36 patients (83.3%). Of these, only 4 patients showed tumor regrowth at the 8th week, compared with the tumor size at the baseline. This suggested a persistent tumor response to this treatment (Figure 3). The median maximum tumor response, i.e., depth of response, was −41.2% (IQR, −23.8–−61.8) in 15 patients (75%) out of 20 responsive patients (Figure 4).

### 2.4. Safety Outcomes

The adverse events (AEs) of any grade observed are shown in Table 2. AEs were observed in all patients, with grade 3 or 4 AEs occurring in 20 (55.6%) patients. The most common AE was liver dysfunction (21 patients, 58.3%) and was mostly grade 2 or lower. Hypertension was the second most common AE (16 patients, 44.4%) and was easily treated with an antihypertensive therapy. Diarrhea, anorexia, and other gastrointestinal symptoms occurred in 15 (41.7%) patients, and hand-foot syndrome was noted in 9 (25.0%) patients. Anorexia and/or malaise was the most common reason for treatment discontinuation and occurred in 13.9% of patients.

## 3. Discussion

A combination of MTAs and ICIs for treating unresectable HCC could represent a promising strategy as these agents act in a complementary manner to induce a tumor response through different antitumor mechanisms. In particular, anti-VEGF agents are ideal for use in combination with ICI treatments as VEGF plays a potential role in establishing an immunosuppressive TME [31,40]. Indeed, the combination of bevacizumab with anti-PD-L1 antibody or lenvatinib with anti-PD-1 antibody has shown considerable antitumor response [32]. However, these combinations have not been used in clinical settings yet. Considering the prolonged activity of these antibodies [33], we can expect similar therapeutic efficacy with lenvatinib and ICI combination therapy when lenvatinib is administered after ICI treatment. This is because anti-PD-1/PD-L1 antibodies administrated prior to lenvatinib might be active even after the initiation of lenvatinib administration.

In this work, we confirmed the efficacy and safety of lenvatinib in patients with HCC who were unresponsive to PD-1/PD-L1 blockade. Approximately 70–80% of patients with HCC are refractory to ICI monotherapy, suggesting the presence of potential resistance mechanisms against these agents during tumor progression [23,24,27]. This justifies an urgent need to establish an effective regimen targeting the antitumor immunity in patients with HCC refractory to ICIs. Our results showed high tumor response rates (ORR: 55.6%; DCR: 86.1%) and a relatively high proportion of patients who sustained a tumor response. Moreover, no rapid, aggressive HCC progression was observed in this study.

The median PFS of 10 months observed in our study was longer than that observed in the lenvatinib cohort of REFLECT trial (7.4 months) [8]. However, it should be noted that the REFLECT trial was a phase III study comparing lenvatinib with sorafenib as a first-line treatment for patients with unresectable HCC (Table 3). However, in our cohort, lenvatinib was used after ICI therapy as a second-, third-, and fourth-line systemic treatment in 22 (61.1%), 9 (25.0%), and 5 (13.9%) patients, respectively. For this reason, the presented data are quite interesting, and they indicate a potential advantage of administering lenvatinib to patients with HCC who have been previously treated with ICIs. In line with this evidence, a recent phase III clinical trial reported a median PFS of 6.8 months (95% CI: 5.7–8.3) with an ORR of 33.2% (95% CI: 28.1–38.6) in patients treated with a combination of atezolizumab and bevacizumab immunotherapy using mRECIST [27]. In addition, a phase I b study that assessed lenvatinib plus pembrolizumab therapy in patients with HCC showed a median PFS of 9.3 months (95% CI: 5.6–9.7) with an ORR of 46.0% (95% CI: 36.0–56.3) using mRECIST [32]. Therefore, lenvatinib administration after ICI therapy showed comparable antitumor response and survival with anti-VEGF agent plus ICI combination therapy (Table 3). It has been reported that anti-PD-1 antibodies can remain bound to CD8^+^ T cells for more than 20 weeks [33]. Therefore, it is quite reasonable to speculate that during lenvatinib treatment, previously administered anti-PD-1/PD-L1 antibody treatment exerts an effect on immune cells as well as tumor cells and induces a synergistic effect with lenvatinib, similar to that observed in the combination therapy of MTAs with ICIs. Further analysis should be conducted in this regard by analyzing the effect of anti-PD-1/PD-L1 antibody on CD8^+^ T cells in patients with HCC.

Overcoming the refractoriness to ICI represents a major challenge to improve the prognosis of patients with cancer including HCC. There are several factors associated with ICI resistance: loss of expression of PD-1/PD-L1 in tumor and antigen-presenting cells [23,41]; low mutation burden limiting tumor associated antigen expression; canonical Wnt/β-catenin pathway activation affecting immune-mediated cell infiltration [42,43,44]; β−2 microglobulin mutations impairing antigen presentation [35,45,46]; and alterations in the janus kinase-signal transducers and activators of transcription pathway affecting interferon-γ signaling [36]. Harding et al. [42] have reported that the activating alterations of Wnt/β-catenin signaling are associated with lower DCR, shorter median PFS, and shorter median OS in patients with HCC treated with ICIs. Furthermore, the expression of multiple repressive immune checkpoint receptors, such as T lymphocyte, including cytotoxic T lymphocyte-associated antigen 4, lymphocyte-activation gene 3, or T-cell immunoglobulin mucin-3, could also be associated with impaired responses to PD-1/PD-L1 blockade.

Another interesting report showed that β-catenin mutation correlates very well with increased expression of FGFR4 [36]. As lenvatinib has a strong inhibitory effect of FGFR4 signaling pathway with IC_50_ to FGFR4 of 43 nM/L, [47] higher response rate (81% vs. 31%) and better PFS (5.5 M vs. 2.5 M) were observed in HCC patients with high FGFR4 expression (positive β-catenin activation) than those with low FGFR4 expression [48]. This might be another explanation of better efficacy of lenvatinib after failure of PD-1/PD-L1 antibody therapy.

Reportedly, VEGF is known as an immunosuppressive growth factor, and its increased level induces the recruitment and activation of immunosuppressive cells expressing VEGF receptors [49]. As VEGF is secreted by HCC cells, it could play a critical role in determining the insufficient immune response noted in patients with HCC. Considering the mutational profile and pathophysiology of HCC, VEGF-associated immunosuppressive TME could be a possible cause of ICI refractoriness in patients with HCC. Indeed, recent data from a phase III clinical trial revealed a higher response rate with anti-VEGF-A and anti-PD-L1 combinatorial therapy than with PD-1/PD-L1 blockade monotherapy. In line with these data, Shigeta et al. reported the occurrence of synergistic antitumor effects of the combination of anti-PD-1 and anti-VEGFR-2 antibodies using murine models of HCC [50]. Combination therapy reprogrammed the TME by increasing CD8^+^ cytotoxic T cell infiltration and activation, shifting the M1/M2 ratio of tumor-associated macrophages with reduction of T regulatory cell and infiltration of C-C chemokine receptor 2-positive monocyte in HCC tissue [50]. From this standpoint, the antitumor mechanism of lenvatinib administered after ICI therapy could be, at least partially, attributed to the targeting of the immunosuppressive cells expressing VEGF receptors and the restoring of the residual anti-PD-1/PD-L1 antibody function of antitumor CD8^+^ cells, in addition to the targeting of tumor cells [51].

There are several anti-VEGF/MTA agents already available for use in clinical settings and for easy use in ICI-based therapy. Lenvatinib administration following anti-PD-1/PD-L1 blockade failure showed stronger antitumor response than that shown by a previously reported ICI monotherapy or lenvatinib monotherapy. Moreover, it showed clinical efficacy similar to that shown by combination therapies. Besides, the safety profile of lenvatinib was generally consistent with that reported in REFLECT trial. Most lenvatinib-related AEs were generally mild and tended to resolve on treatment continuation, without any serious adverse events. This also contributed to the long-term administration and considerable efficacy of lenvatinib.

Despite the promising results, our study has several limitations. First, this was the preliminary retrospective study in nature with no supporting data from comparative studies. Although the number of patients is small, we were able to obtain results that would be a proof of concept, i.e., sequential therapy of ICI-lenvatinib is more effective than lenvatinib alone as an exploratory analysis, so it is warranted to confirm whether our result is correct by conducting prospective randomized study in the future. Second, a selection bias could exist because of the clinical observational nature of the study. Nevertheless, this real-world data analysis indicates high efficacy of lenvatinib therapy following ICI treatment failure, which indicates that lenvatinib can be strongly recommended in cases refractory to ICI monotherapy.

## 4. Materials and Methods

### 4.1. Patients

This single-institute, retrospective study enrolled patients with unresectable advanced HCC who received PD-1/PD-L1 checkpoint blockade followed by lenvatinib from 1 March 2016 to 30 September 2019. Patient profiles are shown in Table 1. A diagnosis of HCC was established using histological findings or radiological modalities according to the American Association for the Study of Liver Diseases criteria [52], Child-Pugh class A liver function, Eastern Cooperative Oncology Group performance status [53] of at most 1, and expected survival time higher than 3 months. mRECIST [39] was used to assess the effect of the treatment to the targeted lesion. Up to 5 target organs were selected, and up to 2 of each organ was used as a target lesions in this study. Spider plots and waterfall plots were constructed using the sum of the longest diameters of maximum two target tumors in the liver.

This study was approved by the institutional review board of the Kindai University Hospital, Ethical Code: 21-42, approved in 2015 and the investigators obtained informed consent from each patient. The written informed consent form was provided from the objectives of the study, along with details of the treatment protocol and the dose of each ICI agent and lenvatinib. The potential risks and benefits of the treatments were also discussed with the patients while obtaining consent.

### 4.2. Treatment Regimens

Lenvatinib (Lenvima; Eisai Co., Ltd., Tokyo, Japan) was orally administered as next-line chemotherapy to patients with unresectable HCC who relapsed or were refractory to previous treatment with anti-PD-1/PD-L1 antibody. The doses of lenvatinib were determined according to body weight: patients weighing <60 kg received 8 mg once daily, whereas those weighing ≥60 kg received an initial dose of 12 mg once daily. Dose reductions or treatment interruptions were considered on a case-by-case basis according to the severity of each AE until symptoms resolved or patient condition returned to baseline.

### 4.3. Efficacy Assessment

Dynamic computed tomography or magnetic resonance imaging was performed every 4–8 weeks, and treatment response was assessed according to mRECIST [39]. ORR was determined by including patients who attained CR and those who attained PR, and the DCR was defined as a sum of ORR and SD. The DCR and ORR were determined with the best response recorded from the beginning of lenvatinib treatment until disease progression or recurrence occurred. For safety assessment, AEs were assessed according to the National Cancer Institute’s Common Terminology Criteria for Adverse Events version 4.0 [54].

### 4.4. Statistical Analysis

Patients were followed up until 29 February 2020 or death using medical records and outpatient visits. The primary outcome was PFS, whereas the secondary outcomes were DCR, ORR, and OS. The baseline characteristics and disease factors of patients are expressed as median IQR. PFS and OS were estimated from the time of the initial lenvatinib administration to the occurrence of tumor progression or death by any cause. Each survival time was measured using the Kaplan–Meier method. Analyses were performed using the SPSS software (version 22; SPSS, Chicago, IL, USA).

## 5. Conclusions

Lenvatinib administered to patients with unresectable HCC who were unresponsive to PD-1/PD-L1 checkpoint blockade demonstrated considerable antitumor activity and a tolerable safety profile. However, because of the limited number of the cases analyzed here, additional investigations with larger cohorts are warranted.

## Figures and Tables

**Figure 1 cancers-12-03048-f001:**
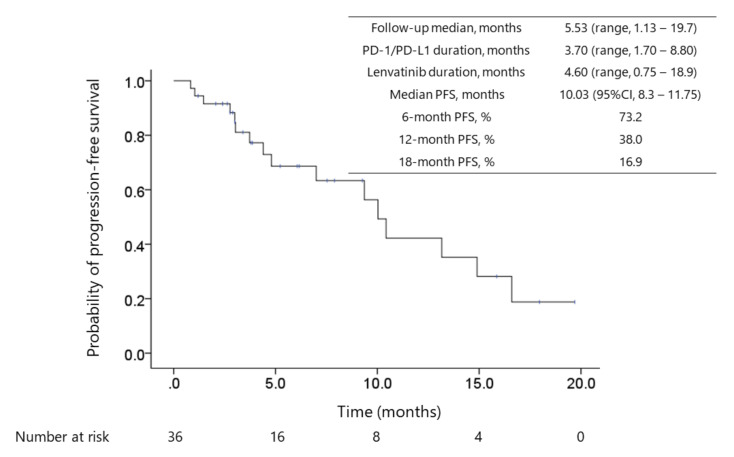
Kaplan–Meier survival curve of progression-free survival (PFS). The median duration of prior PD-1/PD-L1 checkpoint blockade treatment, between ICI termination and lenvatinib treatment initiation, and lenvatinib treatment was 3.7 (IQR, 1.7–8.8) months, 0.95 (IQR, 0.55–4.08) months, and 4.6 (IQR, 2.2–10) months, respectively. Fourteen (39%) patients continued the treatment with complete or partial response status. The median progression-free survival (PFS) was 10 months (95%CI: 8.3–11.8) for 36 patients. Of these, the median PFS was not reached for 22 patients who were administered lenvatinib as the 2nd-line systemic therapy, and the median PFS was 10.4 months (95% CI: 9.08–11.8) for 14 patients who received prior systemic therapy before ICI treatment.

**Figure 2 cancers-12-03048-f002:**
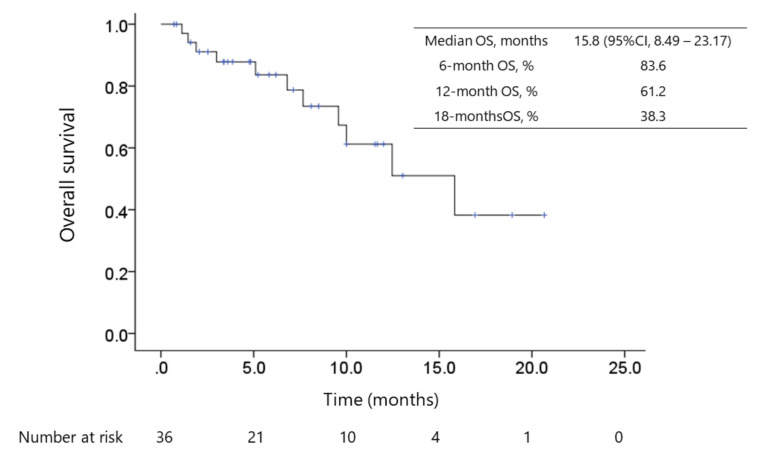
Kaplan–Meier survival curve of overall survival (OS). The median follow-up duration was 5.53 (IQR 2.88–10.38) months and 11 patients died of HCC progression. The median periods of OS since the initiation of lenvatinib therapy and since ICI therapy initiation were 15.8 months (95% CI: 8.5–23.2) and 29.8 months (95% CI: 25.3–34.4), respectively, for 36 patients.

**Figure 3 cancers-12-03048-f003:**
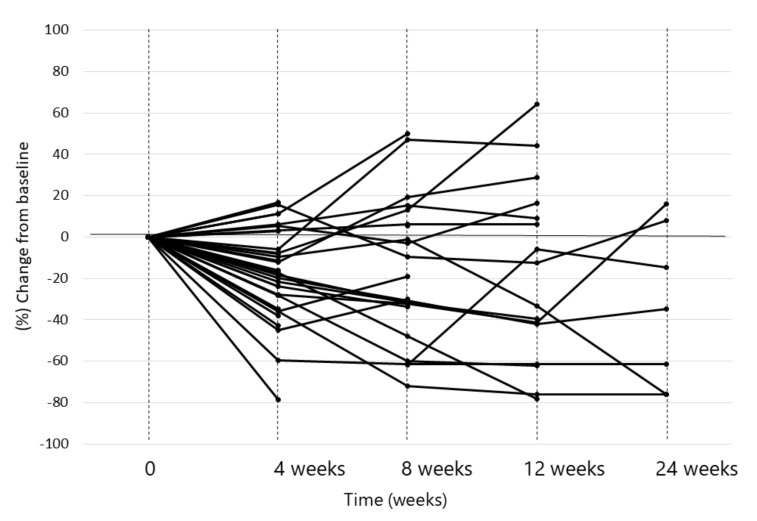
Spider-plot of the HCC cases who were treated with lenvatinib after the failure of immune checkpoint blockade. Decreases of tumor size were observed at the 4th week of the treatment in 30 out of 36 patients (83.3%). Of these, only 4 patients showed increase of tumor size at the 8th week, compared with that at the baseline.

**Figure 4 cancers-12-03048-f004:**
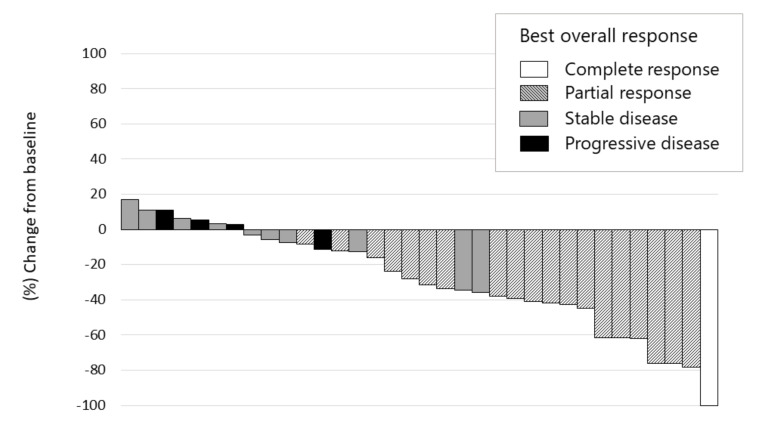
Waterfall plot of maximum tumor response to lenvatinib therapy using mRECIST. Complete response, partial response, stable disease, and progressive disease were noted in 2.8% (*n* = 1), 52.8% (*n* = 19), 30.6% (*n* = 11), and 11.1% (*n* = 4), respectively. The objective response rate and disease control rate in patients who underwent radiological evaluation were 55.6% and 86.1%, respectively, as assessed using the mRECIST. The depth of response was −25.9% (IQR −43.0–−5.2) in all patients and −41.2% (IQR −61.8–−23.8) in responders.

**Table 1 cancers-12-03048-t001:** Baseline characteristics of patients.

Characteristics	Overall (*n* = 36)
Age, median	70 (60.75–76.75)
Gender, male/female	31/5
Etiology: HCV/HBV/NBNC	10/6/20
Albumin, median (g/dL)	3.800 (3.625–4.100)
Total bilirubin, median (mg/dL)	0.600 (0.500–0.975)
ALBI score, median	−2.590 (−2.827–−2.267)
FIB4 index, median	2.778 (1.644–5.191)
BCLC stage, B/C	9/27
Vascular invasion, with/without	5/31
Extrahepatic spread, with/without	26/10
Baseline AFP, median (ng/mL)	76.5 (5.25–11543.5)
DCP, median (mAU/mL)	1303.5 (43.25–7394.3)

HCV, hepatitis C virus; HBV, hepatitis B virus; NBNC, HBV and HCV negative; BCLC, Barcelona Clinic Liver Cancer; ALBI grade, albumin-bilirubin grade; FIB4, The fibrosis-4 (FIB-4) index was calculated as age (year) × AST (IU/L) / (platelet count (109/L) × √ALT (IU/L)) [38]; AST, aspartate aminotransferase; ALT, alanine aminotransferase; AFP, α-fetoprotein concentration; DCP, Des-γ-carboxy prothrombin. We express the inspection value in Median (interquartile range).

**Table 2 cancers-12-03048-t002:** Adverse events observed in this study.

Adverse Events	Any Grade	Grade 3/4
Liver dysfunction	21 (58.3%)	3 (8.3%)
Hypertension	16 (44.4%)	4 (11.1%)
Decreased appetite	15 (41.7%)	1 (2.8%)
Diarrhea	15 (41.7%)	3 (8.3%)
Fatigue	13 (36.1%)	1 (2.8%)
Hypothyroidism	12 (33.3%)	0 (0%)
Hands foots skin reaction	9 (25%)	1 (2.8%)
Hypoalbuminaemia	9 (25%)	1 (2.8%)
Jaundice	8 (22.2%)	2 (5.6%)
Hoarseness	7 (19.4%)	0 (0%)
Thronbocytopenia	6 (16.7%)	3 (8.3%)
Infection	5 (13.9%)	2 (5.6%)
Encephalopathy	3 (8.3%)	1 (2.8%)
Peripheral edema	3 (8.3%)	0 (0%)
Bleeding or haemorrhage	2 (5.6%)	1 (2.8%)
Proteinuria	2 (5.6%)	0 (0%)
Ascites	1 (2.8%)	0 (0%)

**Table 3 cancers-12-03048-t003:** Comparison of anti-tumor response of lenvatinib after PD-1/PD-L1 with that of lenvatinib monotherapy, and anti-VEGF and anti-PD-1 blockade combinations.

Study Name/ID	REFLECT [8]	Presented Study	NCT03006926	NCT03434379
Treatment	Monotherapy	Sequential	Combination	
Agents	Lenvatinib	Lenvatinib after PD-1/PD-L1	Lenvatinib + Pembrolizumab	Atezolizumab + Bevacizumab
Condition				
Design	Phase Ⅲ	Retrospective	Phase Ⅰ b	Phase Ⅲ
Number of patents	478	36	100	336
Setting	1st line	2–4th line	1st line	1st line
Outcome				
ORR (mRECIST)	24.1%	55.6%	46.0%	33.2%
DCR (mRECIST)	73.8%	86.1%	86.0%	72.3%
median PFS	7.4 months	10 months	9.3 months	6.8 months
median OS	13.6 months	15.8 months	22 months	(12 months, OS 67.2%)
		(29.8 months since ICIs started)		
Adverse events	HT 42%	HT 44%	HT 36%	HT 30%
	diarrhea 39%	diarrhea 42%	diarrhea 35%	diarrhea 19%
	appetite loss 34%	appetite loss 42%	fatigue 30%	fatigue 20%
	weight loss 31%	fatigue 36%	AST increase 20%	AST increase 20%
	AST increase 14%	AST increase 58%		
			any grade 99%	
	any grade 99%	any grade 100%	grade 3, 85%	Any grade 98%
	grade 3,4; 75%	grade 3,4; 56%	grade 4, 23%	Grade 3,4; 57%

ORR, objective response rate; DCR, disease control rate; PFS, progression-free survival; OS, overall survival; mRECIST, modified Response Evaluation Criteria in Solid Tumors; ICI, immune checkpoint inhibitor; PD−1, programmed cell death protein 1; PD-L1, PD-ligand 1; HT, hypertension; AST, aspartate aminotransferase. NCT03006926 [32] and NCT03434379 [27].

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
