# Peer review of "Exploratory Analysis of Lenvatinib Therapy in Patients with Unresectable Hepatocellular Carcinoma Who Have Failed Prior PD−1/PD-L1 Checkpoint Blockade"

_cancers, 2020, doi:10.3390/cancers12103048_

Round 1

Reviewer 1 Report

The authors have responded to all concerns.

Reviewer 2 Report

The paper entitled "Exploratory analysis of Lenvatinib
Therapy in Patients with Unresectable Hepatocellular Carcinoma Who Have Failed Prior PD1/PD-L1 Checkpoint Blockade” by Aoki and Kudo, et al. is suitable for publication in the present form without modification.

Reviewer 3 Report

The authors have addressed the criticisms

Reviewer 4 Report

Nice paper. The authors have adequately and successfully addressed the critiques and I have no further points to raise. The Ms can be accepted.

This manuscript is a resubmission of an earlier submission. The following is a list of the peer review reports and author responses from that submission.

Round 1

Reviewer 1 Report

The authors investigate the efficacy of Lenvatinib in HCC patients that no longer respond to PD-1 or PD-L1 targeting therapies. I only have a minor request that the authors re-check some of the sentences (the English is otherwise pretty good). This should hopefully edit out some sentences such as (on page 2) "For this reason, Food and Drug Administration..." should have "the" added. Other than that minor point this seems like a good description of a sound study.

Reviewer 2 Report

The study by Aoki et al., tries to investigate Lenvatinib therapy in HCC patietns who have failed to benefit from PD-1/PD-L1 checkpoint blockade therapy. 

The major concern with this study is the very low number of patients. A patient samples of 36 is considered very low. Furthermore, as mentioned in result 2.2 the treatment regime was discontinued in almost 50% of the participants. The study is still very preliminary and requires a significant patient/treatment population to arrive at a conclusion. 

All the data presented are only presented a numbers and are missing SD or SEM. 

Table 3 comparison clearly points out to the importance of the number of patients as a minimum of 100 patients are required to identify a trend in the treatment regime. It is thus very difficult to draw conclusions in the current study.

Reviewer 3 Report

The paper entitled "Lenvatinib Therapy in Patients with Unresectable Hepatocellular Carcinoma Who Have Failed Prior PD-1/PD-L1 Checkpoint Blockade" by Aoki et al. is suitable for publication in Cancers journal in the present form without further modification. Over all this paper is a clear, concise, and well-written manuscript.

Reviewer 4 Report

The authors provide  an important therapeutic improvement for HCC resistant to ICI.

Authors should give more information on biology of Lenvatinib in introduction.

Authors have administered Lenvatinib after ICI administration. Authors do not discuss why they have chosen to dose after ICI and not simultaneously. Most published studies to date that  evaluate the combination effect of anti-angiogenic agents and ICI give both drugs simultaneously. Lenvatinib in combination with Pembro has become a first standard of care treatment for patients with RCC resistant to ICIs.

Have the authors any data that compares the effect of anti-angiogenic drugs in ICI resistant tumors, administered simultaneously or after ICI?

Authors should discuss some of the signaling pathways that overcome resistance to ICIs when administered  in combination with anti-angiogenic agents.